# Essential Role of BMP4 Signaling in the Avian Ceca in Colorectal Enteric Nervous System Development

**DOI:** 10.3390/ijms242115664

**Published:** 2023-10-27

**Authors:** Tamás Kovács, Viktória Halasy, Csongor Pethő, Emőke Szőcs, Ádám Soós, Dávid Dóra, Pascal de Santa Barbara, Sandrine Faure, Rhian Stavely, Allan M. Goldstein, Nándor Nagy

**Affiliations:** 1Department of Anatomy, Histology and Embryology, Faculty of Medicine, Semmelweis University, 1094 Budapest, Hungary; kovacs.tamas@med.semmelweis-univ.hu (T.K.); szocs.emoke@phd.semmelweis.hu (E.S.); soos.adam@phd.semmelweis.hu (Á.S.); dora.david@med.semmelweis-univ.hu (D.D.); 2PhyMedExp, University of Montpellier, INSERM, CNRS, 34295 Montpellier, France; pascal.de-santa-barbara@inserm.fr (P.d.S.B.); sandrine.faure@inserm.fr (S.F.); 3Pediatric Surgery Research Laboratories, Department of Pediatric Surgery, Massachusetts General Hospital, Harvard Medical School, Boston, MA 02114, USA; rstavely@mgh.harvard.edu (R.S.); amgoldstein@mgb.org (A.M.G.)

**Keywords:** enteric nervous system, neural crest, ceca, hindgut, Hirschsprung disease, BMP4, Noggin, GDNF

## Abstract

The enteric nervous system (ENS) is principally derived from vagal neural crest cells that migrate caudally along the entire length of the gastrointestinal tract, giving rise to neurons and glial cells in two ganglionated plexuses. Incomplete migration of enteric neural crest-derived cells (ENCDC) leads to Hirschsprung disease, a congenital disorder characterized by the absence of enteric ganglia along variable lengths of the colorectum. Our previous work strongly supported the essential role of the avian ceca, present at the junction of the midgut and hindgut, in hindgut ENS development, since ablation of the cecal buds led to incomplete ENCDC colonization of the hindgut. In situ hybridization shows bone morphogenetic protein-4 (BMP4) is highly expressed in the cecal mesenchyme, leading us to hypothesize that cecal BMP4 is required for hindgut ENS development. To test this, we modulated BMP4 activity using embryonic intestinal organ culture techniques and retroviral infection. We show that overexpression or inhibition of BMP4 in the ceca disrupts hindgut ENS development, with GDNF playing an important regulatory role. Our results suggest that these two important signaling pathways are required for normal ENCDC migration and enteric ganglion formation in the developing hindgut ENS.

## 1. Introduction

Neural crest cells (NCC) represent a transient population of multipotent stem cells capable of migrating along specific paths throughout the vertebrate embryo to differentiate into a variety of cell types, including the neurons and glial cells present in the gastrointestinal tract (GI). Neural tube ablation in early chick embryos, and neural tube transplantation experiments between chicken and quail embryos, proved that the NCCs adjacent to somites 1–7 are the major source of enteric glial and neuronal precursors [1,2,3]. A small portion of NCCs from the sacral neural tube level (posterior to the 28th somite) were also found to contribute to the ENS, mostly in the hindgut [4,5,6,7,8]. Once NCCs enter the gut mesenchyme, at which point they are referred to as enteric neural-crest-derived cells (ENCDCs), they migrate along the length of the entire GI tract, to form the enteric nervous system (ENS) [2,3,5,7]. After gut colonization by vagal and sacral ENCDCs, Schwann cell precursors enter the gut alongside the extrinsic nerves [9,10,11,12,13,14,15]. Abnormal ENS formation leads to a number of congenital neurointestinal disorders in humans, including intestinal neuronal dysplasia, hypoganglionosis, and Hirschsprung disease [9,10,13]. Hirschsprung disease is characterized by the absence of the ENS along a variable length of the distal GI tract, resulting in functional obstruction, due to severely impaired gut motility.

Epithelium–mesenchyme recombination experiments demonstrated that the instructive role of the mesenchyme during the ontogeny of the intestine is essential in the formation and patterning of ENCDC-derived enteric plexuses [16,17,18]. For example, mesenchymal expression of BMP4 in the presumptive submucosal area is initiated by sonic hedgehog (SHH) secreted from intestinal epithelial cells and it has multiple effects during ENS development [19,20,21]. Developmental studies have also demonstrated a reciprocal interaction between FGF in the gut epithelium and BMP4 in the mesenchyme, regulating the development of the cecal primordium in mice [22]. This region of the intestine has been proposed to have a key role in hindgut ENS formation. In both avian and rodent embryos, the cecal environment modifies the proliferation, differentiation, and migratory properties of ENCDCs, to promote their journey into the distal hindgut [23,24]. These results suggest that BMP4 produced by the cecal mesenchyme may influence ENCDCs as they colonize the distal gut.

Previous experiments demonstrated that BMP4 acts through BMP receptors on ENCDCs and regulates cell migration [25,26] and differentiation of enteric neurons [27] and glial cells [28]. BMP proteins also play an important role in determining the neuronal-to-glial cell ratio in the ENS, with Noggin-mediated inhibition of BMP increasing the total number of neurons [27], while decreasing the proportion of glia [28]. Transcripts encoding *BMP2* and *BMP4* were each detected in ENCDCs. Similarly, transcripts encoding *NOGGIN* and BMP receptors (*BMPRII*) and their signal transducers (pSMADs) were also detected in the developing ENS, especially after ENCDCs had completed their colonization, suggesting a potential paracrine or autocrine role in ENS development [26,29,30]. BMPs also promote gangliogenesis when ENCDCs aggregate, which is associated with changes in the expression of neural cell adhesion molecule (NCAM) [25,26,31]. Neuron-specific overexpression of Noggin increased the number of neurons in both submucosal and myenteric plexuses of developing rat small intestine [32]. Blocking BMP signaling in the mesenchymal layer of the intestine inhibited the development of smooth muscle and resulted in abnormal patterning of the ENS [26,29,30,33]. However, experimental results from avian and mouse embryos have shown conflicting results. Noggin overexpression in chicken embryos, for example, led to hindgut hypoganglionosis [26], while excess BMP4 in mouse organ cultures inhibited ENCDC migration [25]. BMP4 misexpression in the chicken gizzard mesenchyme leads to hypertrophic ectopically positioned ganglia [25]. BMP2 promotes neuronal differentiation of mouse and rat ENCDCs [34] and also induces the expression of GDNF receptors on the surface of NCCs to promote GDNF responsiveness [35]. Finally, exposure of rat ENCDC cultures to GDNF in combination with high levels of BMP2 or BMP4 leads to significantly more neurons [32].

The present study aimed to characterize the expression of BMP4 during avian hindgut ENS development and to investigate the effect of BMP signaling on ENCDC migration and differentiation. We found that BMP4 is not expressed in the hindgut mesenchyme prior to the arrival of ENCDCs in the ceca. Interestingly, however, BMP4 is expressed in the ceca at least 1 day before ENCDC arrival, leading us to hypothesize that BMP4 may prepare the mesenchymal environment of the ceca ahead of their arrival. Using BMP4-encoding retrovirus and recombinant BMP4 protein, we show that modulating endogenous BMP activity disrupts normal hindgut ENS development. Our data support multiple essential roles for BMP signaling during hindgut ENS formation: (i) BMP4 overexpression leads to large and ectopic enteric ganglia, while its inhibition with Noggin leads to hypoganglionosis; (ii) BMP4 promotes enteric ganglion formation; (iii) GDNF inhibits the response to BMP4, suggesting interactions between these signaling pathways in the regulation of distal ENS formation.

## 2. Results

### 2.1. BMP4 Is An Important Signaling Hub in the Developing Intestine

The avian ceca provide signaling cues essential for ENCDC colonization of the distal hindgut [24]. To explore the major paracrine factors from the ceca that could be driving ENCDC migration, we performed a topological analysis of PPI (protein–protein interaction) networks based on RNA-seq data obtained from 5-day old embryonic (E5) chicken ceca and “interceca”, which represents the segment of intestine between the paired avian ceca (Figure 1A). This approach enables the identification of highly influential regulatory proteins, called hub proteins, based on the number of connections they have (degrees) with other proteins in the network. The number of degrees in this topological analysis was ranked, yielding a list of potential hub proteins with the highest number of predicted interactions (Figure 1B). As we hypothesized that the ceca influences ENCDC migration in a paracrine manner, we specifically focused on the expression of secreted factors. In this analysis, *BMP4*, which was upregulated 1.8-fold in the ceca, ranked the highest for degrees out of all the secreted proteins, making it a strong candidate as an important paracrine signaling hub in ENCDC migration (Figure 1B). To determine whether the ceca and interceca expressed receptors, downstream signaling transducers, and other components of BMP signaling, we evaluated their gene expression through the number of detected transcripts using the number of reads per kilobase per million mapped reads (RPKM) (Figure 1C). From these data, we observed that *BMP4* was not only upregulated in the ceca but also the most highly expressed BMP ligand across the dataset. Receptors for *BMP4* were detected in both the ceca and interceca regions, including ACVR1, ACVR2A, and *BMPRII*.

### 2.2. Expression of BMP4 Signaling Components Supports a Role during Hindgut ENS Formation

*BMP4* expression has previously been described in the vertebrate GI tract [19,20,25,26,30,33,36], but its expression during development of the avian hindgut and formation of the ENS is not known. To examine the spatial pattern of *BMP4* expression in the developing colorectum, whole-mount in situ hybridization was performed at E5 (Figure 2A) and E6 (Figure 2C), shortly after ENCDCs colonize the post-umbilical midgut and ceca, respectively [24]. Undifferentiated ENCDCs were defined as those that express p75 (NGFR, nerve growth factor receptor), SOX10 (markers for ENCDCs and enteric glia), N-cadherin (CDH2), or HNK1. At E5, *BMP4* was restricted to the ceca mesenchyme (Figure 2B), with expression in the hindgut mesenchyme starting at E6 (Figure 2D). Transverse sections at E6 and E7 confirmed that *BMP4* transcripts were specifically present in mesenchyme immediately adjacent to the hindgut epithelium (Figure 2D,E). In contrast to the hindgut, *BMP4* was not expressed in the E5 or E6 midgut mesenchyme (Figure 2A,C). By E14, *BMP4* expression was restricted to the prospective lamina propria (Figure 2F,G) and did not co-localize with p75-expressing ENCDCs [3] or SMA-expressing intestinal smooth muscle (Figure 2F,G–G”).

In situ hybridization was also performed on hindgut sections at various developmental stages using a *BMPRII* riboprobe (Figure 3A,B). In addition, functional BMP activity was confirmed at the same stages with an antibody, recognizing the active and phosphorylated form of SMAD 1, 5, and 8 (namely pSMAD; Figure 3C,D). At E6, *BMPRII* was expressed in the nerve of Remak (Figure 3A,A’). Double staining of E8 hindgut for *BMPRII* transcript (Figure 3B) and p75 protein showed co-expression on ENCDCs (Figure 3B’). To define the relationship of BMP signaling in ENCDCs, immunostaining was performed for pSMAD and N-cadherin on longitudinal sections of E6 hindgut. pSMAD was specifically detected in the ceca mesenchyme (Figure 3C) at E6, with subsequent expression in the N-cadherin+ nerve of Remak, enteric ganglia, inner layer of the muscularis propria, and subepithelial mesenchyme (Figure 3D,D’). As shown in Figure 3C’,C”, pSMAD expression was not present in the N-cadherin+ ENCDCs present at the migratory wavefront. At E6, the ENCDC wavefront was in the ceca, whereas at E8, the ENCDCs had reached the distal hindgut [24]. At this stage, ENCDCs in the enteric ganglia expressed pSMAD (Figure 3D’). To confirm the expression of *BMPRII* and pSMAD in the developing ENS, explanted E6 chick midguts were cultured with GDNF (10 ng/mL) for 24 h, which led to migration of ENCDCs onto the fibronectin coated surface (Figure 3E–F”). *BMPRII* was broadly expressed by the ENCDCs (Figure 3E–E”), whereas pSMAD immunoreactivity was heterogenous (Figure 3F–F”), confirming that ENCDCs possess the cellular machinery for BMP4 signaling.

### 2.3. Inhibition of BMP4 Signaling Leads to Hindgut Hypoganglionosis

Given the observation that the ENS-containing E6 ceca mesenchyme expresses BMP4, whereas the hindgut has neither ENCDC nor BMP4 expression at this stage, we hypothesized that cecal BMP4 is required for hindgut colonization, as previously shown for other morphogens expressed in the ceca, like GDNF, Endothelin-3, and Wnt11 [24]. To test this, the E6 intestine was placed in catenary culture for 48 h in the presence of recombinant BMP4 (200 ng/mL) or Noggin (200 ng/mL) (Figure 4). Tuj1 antibody was used to label neurons and SOX10, to mark all ENCDCs and early enteric glia [3,24]. Culturing E6 intestine for 2 days with no additive (DMEM) led to full ENCDC colonization of the hindgut, with Tuj1+ cells in both enteric plexuses along the length of the hindgut (Figure 4A). In BMP4-treated explants, Tuj1+ cells were present throughout the hindgut and were characterized by large aggregates, suggesting that BMP4 induces premature ganglion formation and hyperganglionosis (Figure 4B). To determine whether BMP4 enhances glial differentiation in addition to premature ganglion formation, enteric neuron specific anti-HU (ELAV-like protein 4) and enteric glia specific brain-fatty acid binding protein (BFABP; FABP7) double immunofluorescence was performed. After culturing E6 intestine for 2 days with no additive, BFABP+ cells were found distributed throughout the nerve of Remak and enteric plexuses (Appendix A). In BMP4-treated cultures, BFABP+ cells were present throughout the intestine in large ganglia, consistent with hyperganglionosis (Figure 4B). Interestingly, the majority of the nerve of Remak cells expressed BFABP, suggesting that BMP4 promotes enteric glial differentiation (Appendix A), as previously reported [28]. In contrast, explants treated with Noggin were characterized by hindgut hypoganglionosis, with fewer Tuj1+ neurons present (Figure 4C). In Noggin-treated cultures, nerve of Remak cells did not express BFABP, suggesting that BMP inhibition inhibits glial differentiation (Appendix A).

The size of enteric ganglia and length of colonized hindgut was also measured in each of the treatment conditions and showed statistically significant differences between the CTRL vs. Noggin treated, CTRL vs. BMP4, and BMP4 vs. Noggin treated groups (Figure 4D,E). Figure 4D shows that Noggin treatment accelerated hindgut colonization. Statistical analysis was performed using Kruskal–Wallis test with a post hoc Dunn’s test, showing significant differences in ganglion sizes between the control vs. BMP4 treated (*p* < 0.01) and BMP4 vs. Noggin treated guts (*p* < 0.001). There was no difference in ganglion diameter between the control and Noggin treated segments (Figure 4E). The opposite effect of BMP4 and Noggin on ganglion formation could result from changes in ENCDC proliferation. Therefore, we examined ENCDC proliferation by measuring EdU incorporation into SOX10+ cells in the presence of BMP4 or Noggin. In controls, EdU+ cells were primarily seen in the mesenchyme and in 20.9±5.7% of SOX10+ cells (Figure 4F). Interestingly, both treatments showed a trend towards decreased ENCDC proliferation, suggesting a potential role for BMP signaling in regulating ENCDC proliferation in the developing gut. For macroscopic analysis, hindgut length (from ileo-cecal junction to the most distal part of the hindgut) and hindgut diameter (middle segments) were measured in each group (*n* = 12), and data were assessed using a Kruskal–Wallis test with a post hoc Dunn’s test. Our quantified results support the morphological differences observed in Figure 4A–C. The average length of the BMP4 treated hindguts (877.3 µm ± 80.23 µm) was significantly (*p* < 0.001) shorter compared to the control (1240 µm ± 100.8 µm) and Noggin (1344 µm ± 81.29 µm) treated groups. Treatment with Noggin further (*p* = 0.02) increased the average length of the hindgut segments compared to the control group. We also assessed the diameter of the explants and did not find significant differences among the groups.

To further test the effect of BMP4 signaling on hindgut ENS development, E6 intestine was cultured for 2 days in the presence or absence of BMP4 or its antagonist Noggin. Normally, the E6 ENCDC wavefront is at the level of ceca [24], and in the distal hindgut by E8 (Figure 5A–C’). Similarly, E6 intestine cultured for 2 days demonstrated complete ENCDC colonization (Figure 4A and Figure 5D–F’). Addition of BMP4 to the culture media led to large and ectopic ganglia formation in the midgut and hindgut (Figure 5G,H,I–I”), with no effect on ganglion size or ENS patterning in the ceca (Figure 5H). By contrast, addition of Noggin caused hindgut hypoganglionosis (Figure 5L,L’). Interestingly, addition of Noggin did not affect ENS formation in the ceca (Figure 5K) but had a significant effect on midgut ENS formation, with complete disruption of normal patterning into two plexuses (Figure 5J). Staining for alpha-smooth muscle actin (SMA) is shown for comparison (Figure 5C,F”,I”,L”). The effect of BMP4 on ENCDC aggregation could result from changes in the mesenchymal compartment of the gut wall; therefore, we determined how the altered BMP signaling affected the radial expression pattern of the extracellular (ECM) proteins known to either support (fibronectin) or inhibit (collagen type III, collagen type VI) ENCDC migration [18]. In E8 hindgut, collagen III and fibronectin were uniformly expressed throughout the hindgut mesenchyme, with intense fibrillar staining under the mesothelial cells and in the subepithelial mesenchymal layer, while the inhibitory collagen type VI was specifically expressed in the inner mesenchymal compartment. Excess of BMP4 or Noggin did not significantly alter the radial expression pattern of ECM (Appendix A).

### 2.4. Retrovirus-Mediated Overexpression of BMP4 Induces Robust Gangliogenesis In Vivo

The effect of BMP4 on the development of the hindgut ENS was also examined in vivo using a replication-competent retrovirus (RCAS) expressing the chicken BMP4 gene (Figure 6). As we previously described [18], RCAS virus was injected into the E6 chicken hindgut mesenchyme and then cultured on an E8 chick chorioallantoic membrane (CAM) for 9 days (Figure 6A). Trophism of avian retroviruses is specific to mesenchymal cells and does not directly target ENCDCs [18,30]. The 3C2 antibody recognizing the RCAS P19 gag protein showed successful and extensive viral replication in the intestinal wall (Figure 6F,F’). As shown in Figure 6G,H, RCAS-BMP4 led to significant enteric hyperganglionosis, with large and disorganized ganglia throughout the gut wall, with loss of the distinct patterning into myenteric and submucosal plexuses observed in the control intestine (Figure 6B–E). Smooth muscle morphology was similarly disturbed (Figure 6I), with no discernible separation into the three muscle layers (muscularis mucosae, circular muscle, longitudinal muscle) normally seen (Figure 6E).

### 2.5. GDNF Inhibits BMP4-Induced ENCDC Aggregation

GDNF signaling is the most important pathway implicated in congenital neurointestinal pathogenesis, and like BMP4, it is spatially and temporally restricted to the cecal region just prior to the arrival of ENCDCs. To test the effect of BMP4 on ENCDCs as they migrate through the cecal region to colonize the hindgut, explanted E6 ceca were cultured with BMP4 in the absence (Figure 7) or presence of GDNF (Figure 8). ENCDCs exhibited robust migration from ceca explants after 48 h of culture in response to GDNF, as previously reported [24], with the majority of ENCDCs expressing the neuronal marker Tuj1 (Figure 7A,A’). When GDNF was removed from the culture media after the first 24 h, and replaced with no additive morphogen, the cell migration over the next 24 h was significantly reduced (Figure 7B) and ENCDCs aggregated into a uniform network of interconnected ganglia (Figure 7B’). When the GDNF was instead replaced after 24 h with either BMP4 (Figure 7C and Appendix A) or Noggin (Figure 7D and Appendix A) for an additional 24 h period, major changes were observed. BMP4 treatment resulted in large ENCDC aggregates with extensive Tuj1 expression (Figure 7D’) and markedly reduced ENCDC proliferation (Figure 7G). In contrast, Noggin prevented normal ganglion formation and disrupted the development of interganglionic connections (Figure 7D,D’). Noggin did not affect ENCDC migration (Figure 7E) or proliferation (Figure 7F) but significantly reduced the rate of neuronal differentiation.

Finally, we assessed the effect of BMP4 treatment in combination with GDNF in ceca explants (Figure 8). When the culture media contained both BMP4 and GDNF, large cell aggregates did not occur (Figure 8B,B’). We therefore conclude that BMP4 acts directly on ENCDCs to reduce their proliferation and promote their aggregation into ganglia, but this effect is normally regulated by the presence of GDNF in the cecal mesenchyme, preventing premature ganglion formation, so that the ENCDC wavefront can continue its migration into the hindgut.

## 3. Discussion

Among developmental anomalies of the neurointestinal system, the most common are the neuronal dysplasias, characterized by ectopic ganglia associated with hypoganglionosis or hyperganglionosis, as well as Hirschsprung disease, which is characterized by colorectal aganglionosis. These congenital diseases are largely due to genetic changes occurring in the GDNF and endothelin-3 signaling pathways [37]. Several lines of evidence support the hypothesis that BMP-4, which, like GDNF and endothelin-3, is also secreted by intestinal mesenchymal cells, contributes to ENS development. However, experiments conducted using a variety of in vivo and in vitro systems and different model organisms have yielded conflicting results [21,23,25,26,32,33]. BMPs (BMP2, 4, and 7) bind and dimerize with type I (BMPR1A or BMPR1B) and type II (BMPRII) receptors to phosphorylate and activate the canonical SMAD1/5/8 signaling cascade [38]. BMP receptors and the BMP antagonists are all expressed in the developing ENS and regulate ENCDC migration [25,26,30] and differentiation into specific neurons and glia [32,34] by controlling the expression of genes involved in ENS specification and maturation. BMP signaling is also involved in the formation and patterning of the enteric ganglia, ensuring the proper distribution and connectivity of enteric neurons. Furthermore, the transcription factor Smad-interacting protein 1 (also known as SIP1 or ZEB2), a negative regulator of BMP4 signaling [39], is involved in the specification, differentiation, and migration of neural crest cells [40], and mutations in this gene are associated with Hirschsprung disease [41,42,43]. Despite the widely recognized relationship between BMP4 and ENS development, the mechanism by which BMP4 controls hindgut colonization is unknown.

Our current study provides several new observations regarding the role of BMP4 in colorectal ENS development: (i) BMP4 expression is restricted to the cecal mesenchyme just prior to the ENCDCs arriving there; (ii) pSMAD is not expressed by wavefront ENCDCs as they migrate through the ceca, which may prevent premature ganglion formation and maintain the migratory stream for hindgut colonization; (iii) presence of excess BMP4 protein led to complete colonization of the hindgut and formation of large, ectopic ganglia, while addition of Noggin to inhibit BMP signaling enhanced ENCDC migration into the hindgut, prevented ganglion formation, and reduced neuronal and glial differentiation; (iv) presence of GDNF reversed the effect of BMP4 on ganglion formation. These results support the idea that ceca mesenchyme rich in BMP4 play an important role in the colorectal colonization by ENCDCs during avian embryo development and its interaction with GDNF.

BMP4 and its receptors have previously been identified during mammalian [25,32] and avian [26,30] ENS development. BMP signaling components have been identified throughout the mesenchyme of the developing chicken gut, except the stomach and hindgut [20,44]. According to Goldstein and his group [26], the earliest BMP4 expression was seen at E8 in subepithelial mesenchyme of the ENCDC colonized hindgut. More recent studies showed that members of the BMP family (BMP2, BMP4, and BMP7) were symmetrically expressed within the E12 chicken midgut and mesentery and BMP signaling, in addition to regulating ENS development, as well as controlling intestinal villus formation, smooth muscle differentiation, and intestinal lopping morphogenesis [29,45,46]. Our transcriptome analysis combined with a whole-mount in situ hybridization study extend previous observations demonstrating that BMP4 is first produced in the ceca mesenchyme at the developmental stage just prior to the colonization by migrating ENCDCs. Early detection of BMP4 expression in E5 chicken ceca is consistent with whole-mount in situ hybridization data observed in 10.5 dpc mouse embryo, where the first sign of BMP4 expression was also seen in the precolonized cecum bud, as soon as the primordium began to grow from the intestine [22].

The ceca are a structure at the junction of the small and large intestines. It has been proposed that exposure to the cecum is indispensable for the colonization of the hindgut by ENCDCs [24]. Expression of developmentally important molecules (GDNF, EDN3, or non-canonical Wnt proteins, including Wnt5a and Wnt11) is higher in the cecum than elsewhere in the hindgut [24]. Remarkably, this is the first site where defects in hindgut colonization are observed in EDN3 and EdnrB mutant mice [47,48,49,50]. At the level of the ceca, the mesenchymal derived EDN3 and Wnt11 growth factors are permissive for the chemoattractive function and inhibitory to the pro-neurogenic effects of GDNF, thereby promoting the migration of undifferentiated ENCDCs through the ceca and into the hindgut [24,51]. Our studies add BMP4 to this signaling complex, supporting a role for BMP4 signaling in colorectal ENS development. In contrast to the ceca, BMP4 expression is turned on in the hindgut mesenchyme after ENCDCs pass the ceca, and it continues to be expressed adjacent to the migrating and differentiating ENCDCs throughout early ENS patterning in the hindgut. This finding is consistent with previous observations that BMP4 signaling influences hindgut ENS development [25,26] and supports the idea that this protein promotes glial and neuronal differentiation and induces cell aggregation, all key steps for ENS formation.

To assess how BMP4 signaling regulates hindgut colonization and supports ENS development, we used virus-mediated overexpression of BMP4 in the precolonized hindgut mesenchyme and further cultured the hindgut on a CAM surface. We used the same avian-specific retroviral expression system in gastrulating chicken embryo as previously reported [30] to induce ectopic and giant ganglion formation in the stomach. These embryos, however, were not viable and died around E5, prior to ENCDC arrival to the ceca. Similarly, in our experiments where RCAS-BMP4 infected embryonic intestine was combined with CAM grafting to extend the culture period, we found ectopic and large ganglia along the in ovo cultured hindgut. This is also consistent with BMP4 inactivation studies, where Noggin misexpression in early chicken embryo led to hypoganglionosis and failure of enteric ganglia formation [26]. The catenary and cell culture results described here also show that Noggin-based inhibition of BMP activity in the hindgut resulted in abnormal ENS development with variable phenotypic expression, including hindgut hypoganglionosis and failure of normal neuronal and glial differentiation. When recombinant Noggin was added to intestinal explants, ENCDC migration was enhanced, ganglion formation was impaired, and enteric neurons were distributed sporadically throughout the hindgut wall, similarly to in previously reported mouse experiments [25]. Contradictory to these findings, misexpression of Noggin during early chicken embryo development produced a significant delay in ENCDC migration and abnormally small ganglia formation [26]. This is consistent with experiments where inhibition of BMP4 activity by Noggin at the level of the neural tube inhibited emigration of neural crest cells [52], reducing the ENCDC pool colonizing the gut. The different effects of Noggin on ENCDC migration between RCAS-Noggin injected early chick embryo and recombinant Noggin protein treated intestine isolated from chicks and mice embryo could be due to different delivery methods.

To determine whether BMP4 can directly influence ENCDC migration and differentiation, we treated precolonized E6 ceca explants with GDNF, which promotes robust migration of ENCDCs out of the gut and onto the fibronectin-coated surface. Twenty-four hours later, after the ENCDCs had emigrated out of the ceca, BMP4 recombinant protein was added to the culture medium, where we found that, consistent with previous results in embryonic mouse gut [53], BMP4 alone did not affect ENCDC migration. Interestingly, BMP4 promoted cell aggregation in this system, similarly to the phenotype observed in our RCAS-BMP4 infected hindguts, while Noggin prevented ENCDC aggregation in vitro, as reported by Chalazonitis and colleagues [32]. In contrast, when BMP4 and GDNF were both added to the ceca explants prior to the onset of cell migration, no ENCDC aggregation occurred. This leads us to hypothesize that during ENS development the presence of high GDNF concentrations in the ceca prevents wavefront ENCDCs from responding to the aggregation-mediating effects of BMP4. This effect must be time-limited, since enteric ganglia will eventually need to form in the ceca. We propose that the migrating cells in the cecum gradually deplete the GDNF reserve. Therefore, while the early arriving ENCDCs are pSMAD-negative and continue to migrate to the proximal hindgut, as the concentration of GDNF in the ceca declines, BMP4 begins to exert its effect on the trailing ENCDC population, which leads to cell aggregation and neuronal differentiation. Our results are consistent with prior studies, where co-administration of BMP factors and GDNF enhanced neuronal differentiation in zebrafish and murine ENS cells [54,55].

In summary, our results show that ENCDC colonization of the hindgut relies on interactions between BMP4 and GDNF signaling in the cecum, which leads to inhibition of premature gangliogenesis prior to hindgut colonization (Figure 9). Accordingly, if BMP4 expression is increased in the hindgut through retroviral misexpression, ectopic ganglia form and hyperganglionosis results. Similarly, when recombinant BMP4 was added to cultured ENCDCs, the cells aggregated into large ganglion-like structures, as opposed to cultures supplemented with both BMP4 and GDNF, where ganglion formation did not occur. Consequently, wavefront ENCDCs reaching the cecum as a response to the chemoattractant effect of GDNF do not differentiate into enteric glia and neurons, and do not form enteric ganglia prematurely, which allows proper post-cecal colonization of the intestine. In addition to highlighting an important aspect of BMP4 signaling during ENS development, this work illustrates how molecular interactions in the cecal mesenchyme are involved in ENCDC colonization of the hindgut.

## 4. Materials and Methods

### 4.1. Embryos

Fertilized white Leghorn chicken (*Gallus gallus domesticus*) eggs were obtained from commercial breeders (Prophyl-BIOVO, Mohács, Hungary) and maintained at 37.5 °C in a humidified incubator. Embryos were staged according to the number of embryonic days (E) or to Hamburger Hamilton (HH) tables [56]. Gut stages were referenced to the chick embryo gut staging table [57] and the ENS formation timetable [3].

### 4.2. RNA-Seq

RNA-seq data from the ceca and interceca of 3 E5 chick embryos were obtained from the GEO database (GSE182783) [1]. This publicly available dataset was re-analyzed focusing on the BMP signaling module. Differential expression of transcripts between the ceca and interceca were determined using the R package edgeR (version 4.2.3) as previously reported with a *p*-value less than 0.01 after FDR correction considering significant differentially expressed genes (DEGs) [58]. Genes with more than 1 read per kilobase of transcript per million mapped reads (RPKM) in 3 samples were considered to be expressed in the dataset. To identify potential signaling hubs in the developing gut, protein–protein interaction networks generated from DEGs were visualized using the web-based tool NetworkAnalyst 3.0 [59] using the STRING functional protein association networks database with a high confidence (800) interaction score threshold. The top 20 (23 due to equal values) genes ranked by degrees (number of potential interactions) in the PPI network topology analysis were considered potentially important signaling hubs. BMP signaling pathway genes were identified from the BMP component of the KEGG database pathway (TGF-beta signaling pathway: hsa04350) with the substitution of equivalent genes in the *Gallus gallus* genome.

### 4.3. Immunohistochemistry

Immunohistochemistry and immunofluorescence techniques were performed on frozen tissue sections and primary cell cultures. For cryosections, tissue was fixed in 4% formaldehyde for 1 h, then infiltrated with 15% sucrose overnight, followed by 7.5% gelatin in 15% sucrose for 1–2 h, then rapidly frozen at −60 °C in isopentane (Merck, 106056, Rowe, NJ, USA). Frozen sections were cut to 12 µm, collected on poly-L-lysine-coated slides (Sigma-Aldrich, P-8920, St. Louis, MO, USA), and stained by immunocytochemistry using the primary antibodies listed in Table 1, as previously described [6]. For obtaining the best result, some primary antibodies needed prior Triton-X 100 (SantaCruz, Dallas, TX, USA, sc-29112) treatment (e.g., SOX10, TUJ1, pSMAD). After primary antibody incubation, sections were incubated with the corresponding biotinylated (Table 2) or Alexa-conjugated fluorescent secondary antibodies (Table 3). In case of chromogenic immunostaining, the development of the signal was with 4-chloro-1-napthol. Section images were recorded using a Nikon Eclipse E800 fluorescence microscope and Zeiss LSM 710 (Carl Zeiss Technika Ltd., Budaors, Hungary) confocal microscope, whole-mount images were recorded using a Nikon SMZ25 (with Prior L200/E unit; Auroscience Ltd., Budapest, Hungary) fluorescence stereomicroscope.

### 4.4. In Situ Hybridization on Sections and Whole-Mount

#### 4.4.1. Whole-Mount In Situ Hybridization

Dissected gastrointestinal tracts were fixed in 4% paraformaldehyde (PFA), dehydrated in methanol, and stored at −20 °C until ready for processing [60,61]. Published chick probes were used: Bmp4 [19,62]. Digoxigenin riboprobe synthesis and whole-mount RNA in situ hybridization were performed as previously described [63,64].

#### 4.4.2. In Situ Hybridization on FFPE Tissue Sections

For sections, GI tissues were fixed in 4% paraformaldehyde at room temperature for 1 h, washed in PBS, gradually dehydrated in ethanol, and embedded in paraffin. Sections (10 µm) were cut using a microtome and collected on poly-L-lysine coated slides. In situ hybridization was performed as previously described [65,66]. All sections were hybridized for 18–24 h. Detection was performed using BM purple, according to manufacturer’s instructions (Roche Molecular Biochemicals, Basel, Switzerland). Digoxigenin riboprobes were prepared as previously described [64] and included probes for BMP4 [19,62] and BMPRII [44].

### 4.5. Intestinal Organ Culture Assay

Guts were removed from E5 (HH26) chick embryos, pinned to silicone-coated tissue culture plates with insect pin as described previously [18], and allowed to float in DMEM (Sigma-Aldrich, D5429) cell culture media containing GDNF or GDNF in combination with BMP4 or Noggin.

### 4.6. Viral Overexpression of BMP4 (+Chorioallantoic Membrane Transplantation)

BMP4 overexpression in the developing chick gut was performed using a replication-competent retroviral vector (RCAS). The RCAS vector used to produce replication-competent retroviruses (RCAS) has been previously described [60,61]. The full-length bmp4 construct [44] was cloned into the shuttle vector Slax as Morgan and Fekete (1996) [67] previously described and then cloned into RCAS using established techniques [68]. The DF-1 chicken fibroblast cell line (ATCC-LGC) was transfected with RCAS-based construct to produce retroviruses expressing BMP4, grown to confluence, and the supernatant was harvested. Viral harvesting, concentration, and titering were performed as described in [69,70]. Approximately 1–5 µL of freshly defrosted virus dyed with fast green was injected per intestine. The gut segment was transplanted onto the chorioallantoic membrane (CAM) of E8 chick embryos. Before transplantation, a small portion of the CAM was gently traumatized by laying a strip of sterile lens paper onto the surface of the epithelium and then removing it immediately. The gut segment was placed over a junction of blood vessels on the CAM and incubated for 9 days. The graft, together with the surrounding CAM was excised, fixed in 4% buffered paraformaldehyde, and embedded in gelatin for further immunohistochemistry. Viral infection was confirmed in all cases using 3C2 immunohistochemistry (1:5) against the viral gag protein. Controls were not injected or injected with empty RCAS vector.

### 4.7. Cell Migration Assay

For ENCDC migration assay, distal midgut without ceca and cecal region was removed from E6 (HH29) chick embryos and cultured on 20 µg/mL fibronectin coated dishes with GDNF (10 ng/mL; R&D Systems, Minneapolis, MN, USA, 212-GD-010; *n* = 12) and GDNF in combination with recombinant BMP4 (200 ng/mL; R&D Systems; 5020-BP) and Noggin (200 ng/mL; R&D Systems, 6997-NG-025).

### 4.8. EdU Labeling

Click-iT Plus EdU AlexaFluor 488 and 647 Imaging Kits (Thermo Fisher, Waltham, MA, USA) were used according to the manufacturer’s instructions to evaluate the cell proliferation rate, either on tissue sections or ENCDCs migrated to tissue culture surface. We developed the fluorescent signals of EdU labeling after immunofluorescent staining had been performed.

### 4.9. Statistics

Data are expressed as mean ± s.d. All statistical analyses were performed using GraphPad Prism version 10.0.2 for Windows, GraphPad Software, Boston, MA USA, www.graphpad.com. After a Shapiro–Wilk normality test, we used a Kruskal–Wallis test with post hoc Dunn’s multiple comparison test of differences. *p* < 0.05 was considered statistically significant.

## Figures and Tables

**Figure 1 ijms-24-15664-f001:**
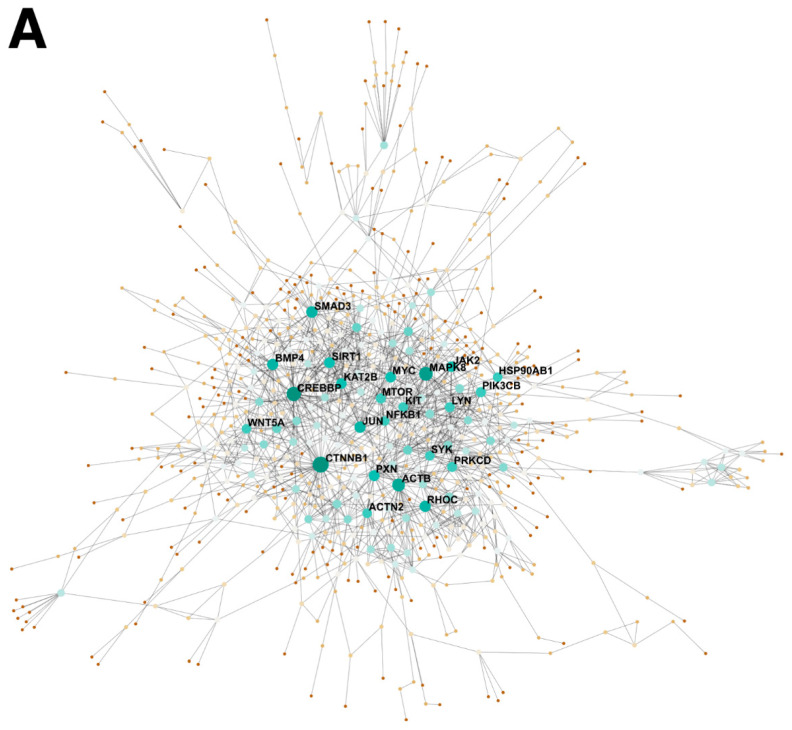
RNA-seq analysis of the BMP signaling network in the E5 ceca and interceca. (**A**) Network topology visualization of PPI networks in differentially expressed genes between the E5 ceca and the interceca (Blue). Node sizes and colors (brown—lowest, green—highest) representing the number of potential interactions between nodes (degrees). (**B**) Heatmap representation of the top signaling hubs ranked by degrees (from PPI topology analysis), with corresponding significance levels by false discovery rate (−Log10FDR) and fold change values (log2FC) for gene expression in the ceca relative to the interceca. (**C**) Total gene expression levels of transcripts presented as reads per kilobase per million mapped reads (Log10RPKM) for ligands, receptors, SMADs, and antagonist in the BMP pathway in the ceca and interceca. Corresponding significance levels (−Log10FDR) and fold change (log2FC) values for differential gene expression in the ceca relative to the interceca are provided.

**Figure 2 ijms-24-15664-f002:**
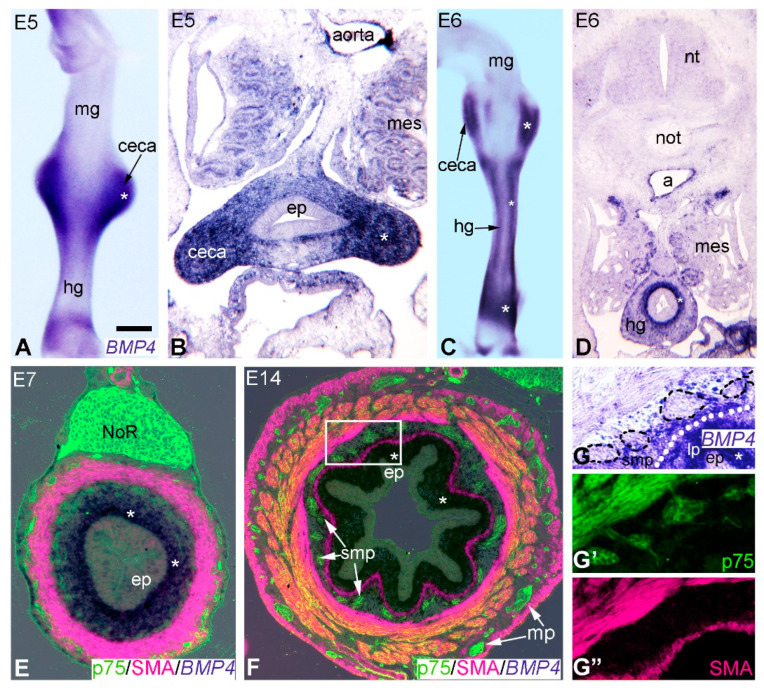
Expression of *BMP4* during hindgut development. Whole-mount in situ hybridization of an E5 gut shows *BMP4* expression primarily in the ceca (**A**–**C**). From E6 through E14, *BMP4* is expressed in the inner mesenchyme along the entire hindgut and ceca, shown in transverse sections (**B**,**D**,**E**,**F**). Double staining of E7 and E14 hindgut using BMP4 in situ (**E**,**F**) and p75 immunofluorescence (**E**,**F**,**G**–**G”**) shows that *BMP4* is not expressed by the p75+ ENCDCs Boxed are in (**F**) is magnified in (**G**–**G”**). Dotted lines denote location of lamina muscularis mucosae. Asterisks mark high expression territory of *BMP4* in (**A**–**G**). Scale bar: 350 μm (**A**), 175 μm (**B**), 700 μm (**C**), 500 μm (**D**), 100 μm (**E**), 120 μm (**F**); 50 μm (**G**–**G**”). a, aorta; ep, epithelium; hg, hindgut; lp, lamina propria; mes, mesonephros; mg, midgut; mp, myenteric plexus; NoR, nerve of Remak; nt, neural tube; not, notochord; smp, submucosal plexus.

**Figure 3 ijms-24-15664-f003:**
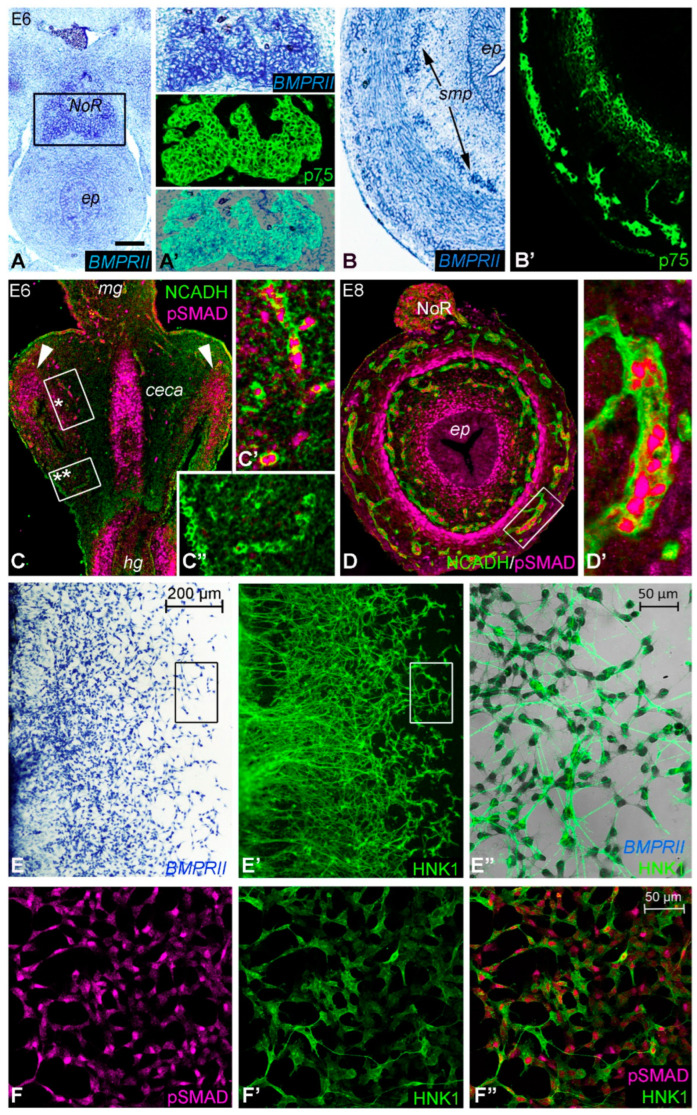
Expression of *BMPRII* and phosphoSMAD during hindgut development. In situ hybridization of an E6 gut shows *BMPRII* expression concentrated in the nerve of Remak (**A**). Boxed area in (**A**) is magnified in (**A’**). At E8, *BMPRII* was expressed in the p75+ ENS (**B**,**B’**). Longitudinal section through the ceca and hindgut at E6 shows N-cadherin+ (NCADH) ENCDCs throughout the ceca mesenchyme. N-cadherin is a typical adhesion molecule for all migrating neural crest cells (**C**). Arrowheads indicate localized pSMAD expression in ceca mesenchyme. Boxed areas marked with asterisks are magnified in (**C’**,**C”**). Double-immunofluorescence with the pSMAD antibody shows that the trailing cells shown in figure (**C’**) co-expressed pSMAD, while the wavefront cells (**C”**) defined as the distalmost N-cadherin expressing undifferentiated ENCDCs, showed no expression of pSMAD. At E8, pSMAD was strongly expressed by the subepithelial mesenchyme, the inner layer of the tunica muscularis, as well as in the developing N-cadherin immunoreactive ENS (**D**); boxed area in (**D**) is magnified in (**D’**). Explanted E6 chick midgut was cultured with GDNF, which induces robust ENCDC migration from the intestinal tissue. Both *BMPRII* (**E**) and pSMAD (**F**) are strongly expressed by the migrating HNK1+ ENCDCs (**E’**,**F’**,**F”**). Boxed area in (**E**,**E’**) is magnified in (**E”**). Scale bar: 120 μm (**A**), 60 μm (**A’**), 50 μm (**B**,**B’**), 180 μm (**C**), 25 μm (**C’**,**C”**), 85 μm (**D**), 20 μm (**D’**), 40 μm (**F**–**F”**). ep, epithelium; hg, hindgut; mg, midgut; NoR, nerve of Remak; smp, submucosal plexus.

**Figure 4 ijms-24-15664-f004:**
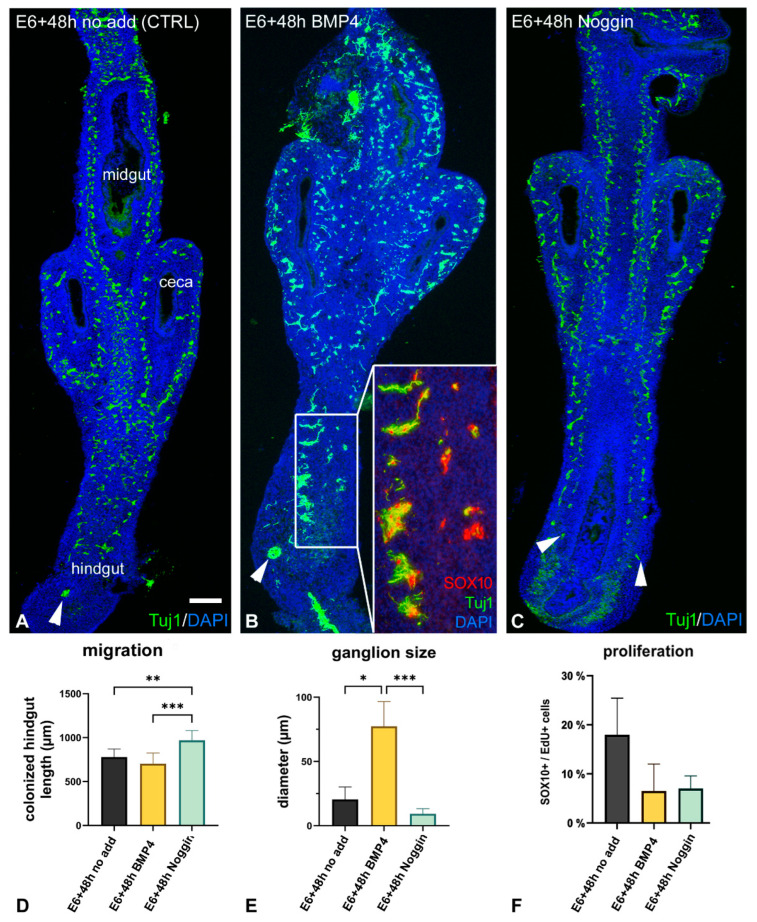
BMP4 signaling is required for hindgut colonization. E6 chick gut was cultured in catenary culture for 2 days in the absence of additives (**A**), with BMP4 protein (**B**), or with Noggin protein (**C**). Longitudinal sections are shown in panels (**A**–**C**). Arrowheads indicate wavefront cells. Corresponding transverse section through the mid-hindgut used for quantification is shown beneath (**E**). Addition of BMP4 induced large ganglion formation in the hindgut (**B**, and inset in (**B**)), whereas inhibition of BMP4 signaling with Noggin led to small ganglia formation (**C**). *n* = 21. Cells in the ganglion expressed neuronal markers Tuj1, and SOX10, which labels ENCDCs and enteric glial cells (**B**, and inset in (**B**)). Quantification of the length of SOX10+ ENCDC colonization of the hindgut from the ceca (*n* = 7 guts per group) (**D**). The average diameter of the SOX10+/TUJ1+ enteric ganglia in control DMEM-treated hindgut, compared to BMP4-treated and Noggin-treated hindgut ganglia. Data are mean ± s.d. *n* = 9 (3 different areas were measured on 3 hindguts/group) (**E**). Following BMP4 and Noggin treatment, there was a tendency toward reduced ENCDC proliferation compared with no additive controls (**F**). Kruskal–Wallis with Dunn’s multiple comparison test was used for statistical analysis. *n* = 12. *** *p* < 0.001, ** *p* < 0.01, * *p* < 0.05. Scale bar: 350 μm (**A**–**C**), 60 μm (**B** inset).

**Figure 5 ijms-24-15664-f005:**
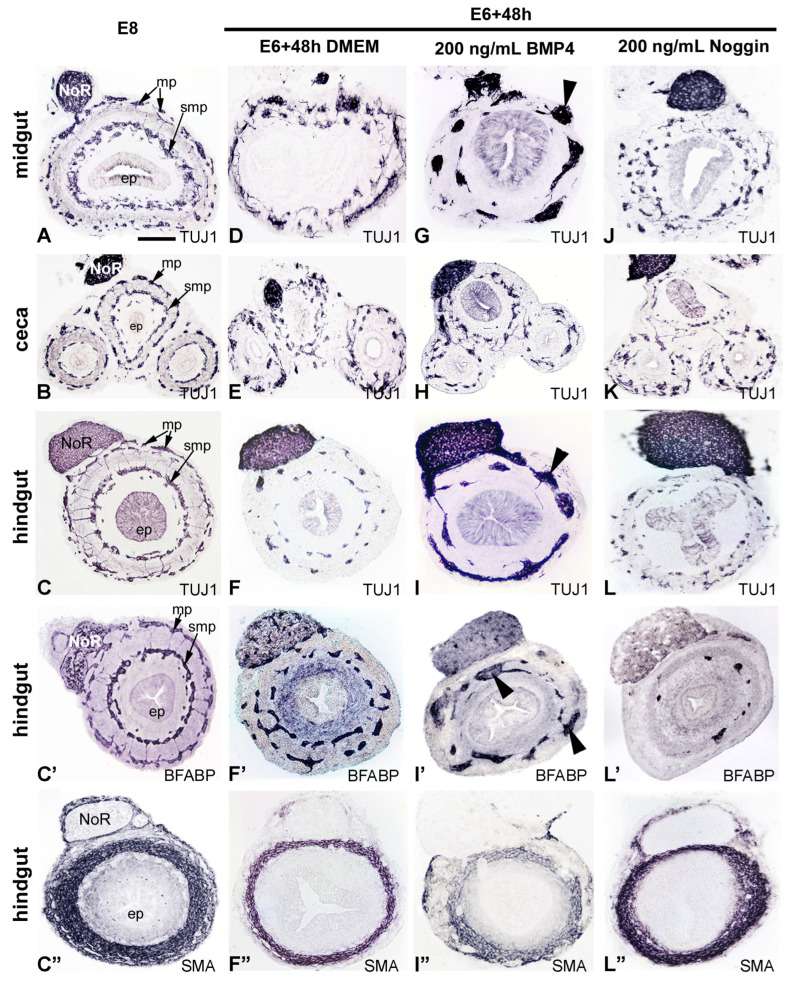
BMP4 overexpression led to hyperganglionosis in chicken midgut and hindgut. Cross section of the E8 midgut (**A**), ceca (**B**) and hindgut (**C**) immunostained with anti-Tuj1 (βIII-tubulin) (**C**), anti-BFABP (**C’**) and anti-alpha-SMA (**C”**) monoclonal antibodies. Chick E6 gut was cultured in catenary culture for 2 days in the absence of additives (**D**–**F”**), with BMP4 protein (**G**–**I”**), or with Noggin (**J**–**L”**). Transverse sections are shown through the mid-hindgut. Addition of BMP4 induced ectopic and large ganglia in the midgut ((**G**), arrowhead) and hindgut ((**I**,**I”**), arrowhead), whereas inhibition of BMP4 signaling led to significant hindgut hypoganglionosis (**L**,**L’**). Noggin treatment induced the disorganization of the midgut ENS: (**J**) compared to control (**D**) and E6 gut cultured for 48 h. Interestingly, BMP4 treatment did not significantly alter the ENS pattern at the level of the ceca, as seen by comparing the ENS morphology of Tuj1 immunostained sections of BMP4 treated ceca (**H**) to control (**B**) and E6 gut cultured for 48 h (**E**). Consecutive cross-sections of the hindgut show the presence of Tuj1+ enteric neuron (**C**,**F**,**I**,**L**), BFABP+ enteric glia cells (**C’**,**F’**,**I’**,**L’**) and alpha-SMA+ smooth muscle layers (**C”**,**F”**,**I”**,**L”**). Scale bar in A: 230 μm (**A**,**D**,**G**,**J**), 400 μm (**B**,**E**,**H**,**K**), 360 μm (**C**–**C”**,**F**–**F”**,**I**–**I”**,**L**–**L”**). ep, epithelium; mp, myenteric plexus; NoR, nerve of Remak; smp, submucosal plexus.

**Figure 6 ijms-24-15664-f006:**
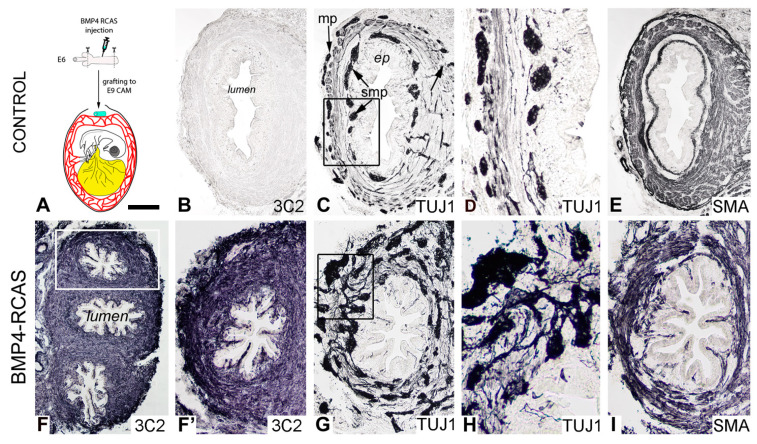
Retrovirus-mediated misexpression of BMP4 disrupted hindgut ENS development. Overexpression of BMP4 induced ectopic and large ganglia in the gut wall. Schematic illustration demonstrates ex vivo injection of dissected E6 hindguts with RCAS-BMP4 and cultured for 9 days on an E9 CAM (**A**). Immunostaining of consecutive sections obtained from control CAM grafts for 3C2 antibody specific for RCAS (**B**), Tuj1 ((**C**), boxed area magnified in (**D**)) and alpha-SMA (**E**) is shown for comparison. RCAS was strongly expressed throughout the gut wall ((**F**), boxed area magnified in (**F’**)), and associated with hyperganglionosis, abnormal radial patterning of enteric ganglia ((**G**), boxed area magnified in (**H**)) and abnormal smooth muscle development (**I**). The scale bar is on the figure: 250 μm (**B**,**C**,**E**,**I**), 80 μm (**D**), 560 μm (**F**), 280 μm (**F’**,**G**), 140 μm (**H**). ep, epithelium; mp, myenteric plexus; smp, submucosal plexus.

**Figure 7 ijms-24-15664-f007:**
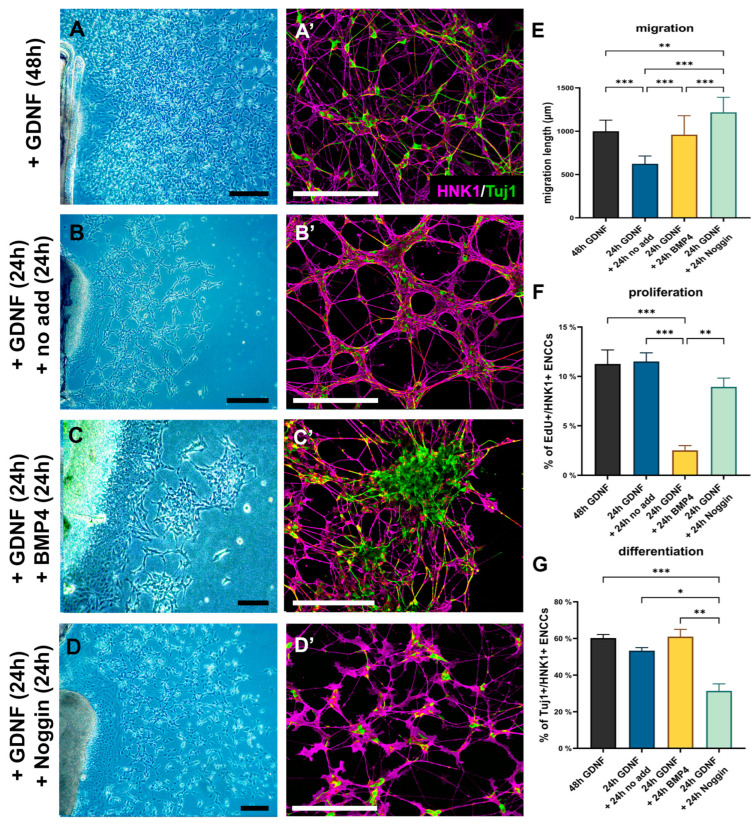
BMP4 induces in vitro gangliogenesis. E6 ceca were cultured on fibronectin-coated surface with GDNF for 48 h (**A**) or after 24 h the GDNF was removed, followed by 24 h with either no added factors (**B**) or addition of BMP4 (**C**) and Noggin (**D**) protein. Cell cultures were stained with HNK1 and Tuj1 antibodies (**A’**–**D’**) to assess ENCDC migration distance and neuronal differentiation. (**E**) Addition of BMP4 did not significantly alter the migration compared to 48 h GDNF cultures, but Noggin increased the migratory distance of the ENCDCs. (**F**) The addition of BMP4 protein significantly inhibited the proliferation of cultured ENCDCs. (**G**) There was no difference in percentage of Tuj1+/HNK1+ cells between (48 h) GDNF and 24 h GDNF + 24 h no additive. Excess of BMP4 did not induced significant neuronal differentiation of ENCDCs but initiated their aggregation. In contrast, in the presence of Noggin, neuronal differentiation and aggregation were markedly reduced. All scale bars represent 200 µm. *n* = 7–10 cell cultures/experiment. Kruskal–Wallis with Dunn’s multiple comparison test was used for statistical analysis. *** *p* < 0.001, ** *p* < 0.01, * *p* < 0.05.

**Figure 8 ijms-24-15664-f008:**
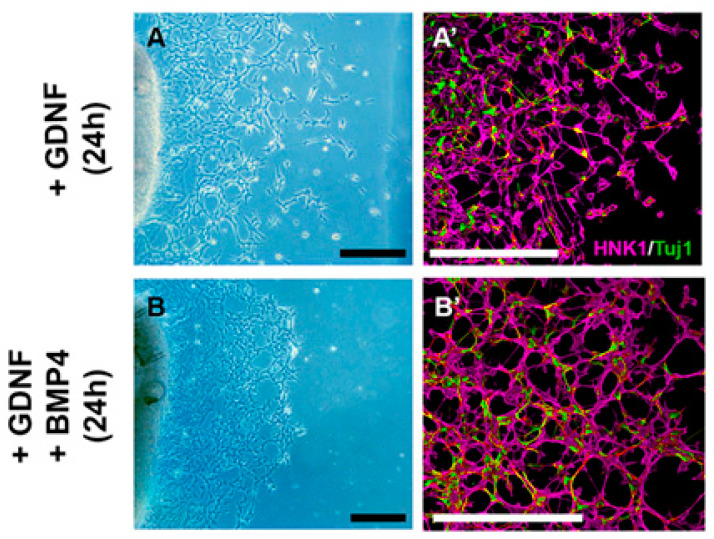
Concurrent administration of GDNF and BMP4 hinders the ganglion-forming effect of BMP4. E6 ceca were cultured on fibronectin-coated surface with GDNF for 24 h (**A**) or GDNF+BMP4 (**B**) for 24 h. Cell cultures were stained with HNK1 and Tuj1 antibodies (**A’**,**B’**) to assess ganglion formation. GDNF-mediated ENCDC migration from E6 ceca was robust at 24 h, with the distance of cell migration shown (**A**). There was no significant difference between the GDNF and GDNF+BMP4 groups, but the simultaneous administration of GDNF with BMP4 inhibited aggregation of Tuj1+/HNK1+ cells. *n* = 7 cultures/experiments. All scale bars represent 200 µm.

**Figure 9 ijms-24-15664-f009:**
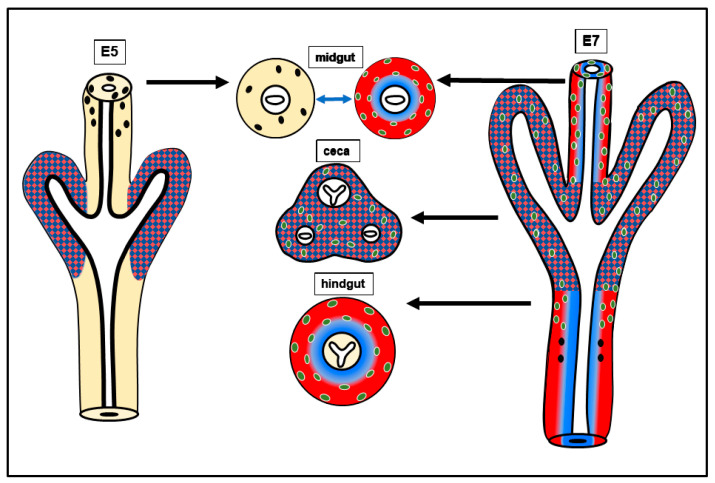
Model of hindgut ENCDC colonization and the roles of BMP4. BMP4 reduces ENCDC proliferation, promotes differentiation, and induce gangliogenesis. GDNF signaling promotes ENCDC proliferation, migration, and differentiation. Expression of both growth factors is precisely localized to the cecal mesenchyme before the arrival of ENCDCs, which suggests that these molecules act locally to directly or indirectly influence the migration of ENCDCs. We hypothesize the overlapping cecal expression of BMP4 and GDNF is required for hindgut ENS formation: GDNF inhibits BMP4 from promoting gangliogenesis in the cecum, allowing wavefront cells to proceed to the hindgut and continue their journey in a BMP4-free, GDNF-rich outer mesenchyme, to complete ENS formation in the colorectum. Undifferentiated ENCDCs and wavefront cells (black), differentiated ENS cells (green); mesenchymal cells (yellow); GDNF expression territory (red); BMP4 expression territory (blue).

**Table 1 ijms-24-15664-t001:** Primary antibodies used.

Antibody (Clone)	Host Species	Antigen Specificity	Dilution	Vendor (Catalog Number)
HNK-1	mouse	IgM	1:50	Thermo Fisher, Waltham, MA, USA (MA5-11605)
Tuj1 (AA10) ^1^	mouse	IgG2a	1:100	Santa Cruz, Dallas, TX, USA (sc-80016)
SMA (1A4)	mouse	IgG2a	1:400	Dako, Santa Clara, CA, USA (M0851)
p75^NTR^	rabbit (polyclonal)	IgG (H + L)	1:300	Promega, Madison, WI, USA (G3231)
pSMAD1/5/9 ^1^	rabbit (polyclonal)	IgG (H + L)	1:200	Cell Signaling, Danvers, MA, USA (#13820)
Bfabp	rabbit (polyclonal)	IgG (H + L)	1:50	kind gift of Dr. Thomas Müller
HuC/D (16A11)	mouse	IgG2b	1:100	Invitrogen, Waltham, MA, USA (A-21271)
N-cadherin (GC-4)	mouse	IgG1	1:200	Sigma-Aldrich, St. Louis, MO, USA (C3865)
N-cadherin (6B3-c)	mouse	IgG1	1:5	DSHB
SOX10 ^1^	mouse	IgG1	1:200	Santa Cruz, Dallas, TX, USA (sc-365692)
RCAS gag protein (AMV-3C2)	mouse	IgG1	1:5	DSHB
collagen III. (3B2)	mouse	IgG1	supernatant	DSHB
collagen VI. (39)	mouse	IgG1	supernatant	DSHB
fibronectin (B3/D6)	mouse	IgG2A	1:3	DSHB

^1^ This target required 0.1% Triton-X 100 permeabilization prior to labeling with primary antibodies.

**Table 2 ijms-24-15664-t002:** Biotinylated secondary antibodies used.

Host	Target	Specificity	Vendor (Catalog Number)
horse	Anti-Mouse	IgG (H + L)	Vector Laboratories, Newark, CA, USA BA-2000
goat	Anti-Rabbit	IgG (H + L)	Vector Laboratories BA-1000
goat	Anti-Mouse	IgM (µ chain)	Vector Laboratories BA-2020 ^1^

^1^ All biotin conjugated secondary antibodies were diluted 1:200 in PBS-BSA prior to use.

**Table 3 ijms-24-15664-t003:** Fluorescent secondary antibodies used.

Host	Target	Specificity	Excitation Wavelength	Vendor ^1^ (Catalog Number)
donkey	Anti-Mouse	IgG (H + L)	488 nm	A21202
donkey	Anti-Mouse	IgG (H + L)	594 nm	A21203
donkey	Anti-Mouse	IgG (H + L)	647 nm	A31571
donkey	Anti-Rabbit	IgG (H + L)	488 nm	A21206
goat	Anti-Mouse	IgG1	488 nm	A21121
goat	Anti-Mouse	IgG1	594 nm	A21125
goat	Anti-Mouse	IgG2a	488 nm	A21131
goat	Anti-Mouse	IgG2a	594 nm	A21135
goat	Anti-Mouse	IgG2b	594 nm	A21145
goat	Anti-Mouse	IgM	594 nm	A21044
goat	Anti-Mouse	IgM	633 nm	A21046

^1^ All fluorochrome conjugated secondary antibodies were purchased from Invitrogen-Life Technologies, Waltham, MA, USA and diluted 1:200 in PBS prior to use.

## Data Availability

Data are available in a publicly accessible repository that does not issue DOIs. Publicly available datasets were analyzed in this study. The data can be found here: Gene Expression Omnibus database under accession number GSE182783. https://www.ncbi.nlm.nih.gov/geo/query/acc.cgi?acc=GSE182783 (accessed on 31 January 2022).

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
