# Peer review of "Essential Role of BMP4 Signaling in the Avian Ceca in Colorectal Enteric Nervous System Development"

_ijms, 2023, doi:10.3390/ijms242115664_

Round 1

Reviewer 1 Report

Comments and Suggestions for Authors

Comments to the authors

This is an interesting study that carefully dissects the role of BMP signaling in hindgut enteric nervous system development using an avian model system. This is important work to determine the role of BMP signaling in addition to ret/GDNF and Endothelin signaling. However, I have different comments regarding the presentation of data and some of the experimental set ups that need to be addressed.

Major comments

1)      Generally, boxes that indicate close-ups often are not in the actual shape of the close-up (for example Fig. 1 G, G’ and G’’) – this is very confusing and needs to be changed throughout the figures

2)      Generally, boxes that indicate close-ups often are not in the actual shape of the close-up (for example Fig. 1 G, G’ and G’’) – this is very confusing and needs to be changed throughout the figures

3)      The authors perform an extensive expression analysis of the BMP signaling pathway components. I have several points that would clarify where the BMP signaling pathways are expressed

a.       Figure 1A/B – the network is not readable because it’s way too small. Please find a better representation so the nodes can actually be seen

b.       Figure C-D – it’s a bit hard to understand what is actually presented here – please provide more extensive description in the figure legend and material & methods – for C, what information should the reader get from this figure – these interaction partners are not relevant for the remaining paper, so it’s unclear why this information is there. Same is true for D – not clear what the information is for.

c.       Figure E: what does LogFC and LogFDR mean and what information does it provide

4)      Expression analysis

a.       The expression analysis would benefit from a summary at the end where the BMP signaling components are expressed – it is quite hard to follow the expression analysis description over time, so summarizing it would be helpful. Also, adding arrows or outline where you see expression would help understanding of expression especially for colometric in situ images.

b.       Lines 129/130: The authors mention markers that they don’t use in the analysis – please remove or explain relevance

c.       Fig. 2E – the label ep is basically where the lumen is – please place label appropriately.

d.       Fig. 3A, A’ – A shows just BMPRII expression, but A’ shows two channels, where is p75 in A?

e.       Fig. 3C, what’s NCADH and what does it label?

f.        Line 159-161: this information comes from later work, but cannot really be deduced from expression analysis and should be removed here

5)      Inhibition of BMP signaling

a.       This experiment modifies BMP signaling throughout the gut, not only in the ENS. Are there changes in gut size, length, thickness? Are these accounted for? In the discussion the authors mention that BMP signaling impacts gut epithelium and smooth muscle cell development (which you also see in Figure 5 – the authors do not write what the staining is for, please add that information) – as these tissues signal each other during development, how do the authors know that their data is due to effects within the ENS and not due to indirect effects? Indirect effects would still be interesting but the authors have to address this in the paper.

b.        Lines 190-192: does Sox10 at the stage not also label ENS glia?

c.       Figure 4E – is there a difference between no add & Noggin treated?

d.       Figure 4F, how do BMP and Noggin treatments both reduce proliferation? How is this explained? Normally opposite treatments do not have the same effect.

e.       Figure 6 – not clear if F and F’ is 3C2 staining? Please explain which cells receive the virus

f.        Figure 7 E-G: consider using letter code to indicate which column is significant from which – the bars are confusing

6)      Please refer to Fig 9 in the discussion.

7)      Figure 9 is hard to understand – for example, why is there an arrow going from E5 to the middle cross section of the ceca, when there are no enteric ganglia present. Is there a different color for migrating progenitor cells vs enteric ganglia? Why should the reader compare the middle-section between E5 & E7 for the midgut?

Minor comments

-          Some of the figures use red and green color coding which is not suitable for color-blind people – please use green-magenta or other color palettes suitable for color-blind people.

Comments on the Quality of English Language

no comments

Author Response

Reviewer 1

We greatly appreciate our reviewers’ thoughtful and constructive comments.  We are also delighted that all two reviewers recognized the value of this work with very positive comments.

Major comments

1)      Generally, boxes that indicate close-ups often are not in the actual shape of the close-up (for example Fig. 1 G, G’ and G’’) – this is very confusing and needs to be changed throughout the figures 2)      Generally, boxes that indicate close-ups often are not in the actual shape of the close-up (for example Fig. 1 G, G’ and G’’) – this is very confusing and needs to be changed throughout the figures

Re: Thank you for the observations. Inset images were rotated for better representation, we have corrected the shape of the marked area in image F.

3)      The authors perform an extensive expression analysis of the BMP signaling pathway components. I have several points that would clarify where the BMP signaling pathways are expressed

  1. Figure 1A/B – the network is not readable because it’s way too small. Please find a better representation so the nodes can actually be seen

We thank the Reviewer for the suggestion, we removed Figure 1A, which makes the network better understandable.

  1. Figure C-D – it’s a bit hard to understand what is actually presented here – please provide more extensive description in the figure legend and material & methods – for C, what information should the reader get from this figure – these interaction partners are not relevant for the remaining paper, so it’s unclear why this information is there. Same is true for D – not clear what the information is for.

Detailed description for information presented in Figure 1C was added to the figure legend and main text. We agree with the reviewer’s comment that Figure 1D did not add any relevant information, therefore we removed it from the panel.

  1. Figure E: what does LogFC and LogFDR mean and what information does it provide

LogFC provides information on the fold change values for gene expressions in the ceca relative to the interceca, whereas LogFDR is the measure of significance levels by false discovery rate. Terms were explained in the figure legend.

4)      Expression analysis

  1. The expression analysis would benefit from a summary at the end where the BMP signaling components are expressed – it is quite hard to follow the expression analysis description over time, so summarizing it would be helpful. Also, adding arrows or outline where you see expression would help understanding of expression especially for colometric in situ images.

Thank you for the suggestion, structures were labeled as suggested, we added asterisks to the figures to highlight the expression territory of BMP4 gene.

  1. Lines 129/130: The authors mention markers that they don’t use in the analysis – please remove or explain relevance

Thank you for the observation. We aimed to highlight the markers for detecting ENCDCs, which are used throughout the results section.

  1. Fig. 2E – the label ep is basically where the lumen is – please place label appropriately.

Thank you for pointing this out, the label was repositioned.

  1. Fig. 3A, A’ – A shows just BMPRII expression, but A’ shows two channels, where is p75 in A?

In Fig. 3A BMPRII expression is shown without the green fluorescent channel for better visibility. In the E6 hindgut, only the nerve of Remak expresses the p75 antigen (together with BMPRII), as shown magnified in Fig. 3A’.

  1. Fig. 3C, what’s NCADH and what does it label?

Thank you for drawing our attention to the inaccuracy of the description in the figure legend. N-cadherin is a typical adhesion molecule for all migrating neural crest cells, this information was added to the figure legend.

  1. Line 159-161: this information comes from later work, but cannot really be deduced from expression analysis and should be removed here

We thank the Reviewer for the observation; the section was removed as suggested.

5)      Inhibition of BMP signaling

  1. This experiment modifies BMP signaling throughout the gut, not only in the ENS. Are there changes in gut size, length, thickness? Are these accounted for? In the discussion the authors mention that BMP signaling impacts gut epithelium and smooth muscle cell development (which you also see in Figure 5 – the authors do not write what the staining is for, please add that information) – as these tissues signal each other during development, how do the authors know that their data is due to effects within the ENS and not due to indirect effects? Indirect effects would still be interesting but the authors have to address this in the paper.

We appreciate these valuable comments and suggestion. Although BMP4 has a major influence on ENS development, the presence of BMP4 receptors expression on mesenchymal cells indicates that the effect is mediated both directly influencing the ENCDCs and indirectly via alterations in the gut microenvironment. To determine whether BMP4 has a direct effect on ENCDC development, E6 chicken ceca was cultured with BMP4 or Noggin (Fig. 7). As expected, BMP4 led to the aggregation of ENCDCs (Fig. 7A), and Noggin inhibited this effect. We also assessed the indirect effect of BMP4/Noggin signaling in 48 h cultured hindgut explants by analyzing the expression pattern of extracellular (ECM) proteins known to inhibit or support ENCDCs migration: collagen III, collagen VI, and fibronectin, respectively. In E8 hindgut, collagen III and fibronectin are uniformly expressed throughout the hindgut mesenchyme, with intense fibrillar staining in the subepithelial mesenchymal layer, while the inhibitory collagen type VI is specifically expressed in the inner mesenchymal compartment (19). Excess of BMP4 or Noggin did not significantly alter the radial expression pattern of ECM (Supplemental Fig 2). The Results section has been modified accordingly (see the highlighted paragraph) and a new supplemental figure (Supplemental Figure 2) is added.

Regarding to the labeling of Fig. 5, the suggested changes have been made.

  1. Lines 190-192: does Sox10 at the stage not also label ENS glia?

Sox10 at this stage labels all ENCDCs and early enteric glia, this information was added to the main text, along with references regarding the immunophenotype of differentiating avian ENCDCs during colorectal ENS formation.

  1. Figure 4E – is there a difference between no add & Noggin treated?

Differences in ganglion size presented in Fig. 4E are not significant between no add and Noggin treated groups, adjusted p-value = 0.34.

  1. Figure 4F, how do BMP and Noggin treatments both reduce proliferation? How is this explained? Normally opposite treatments do not have the same effect.

We greatly appreciate the Reviewer pointing out this issue.  In our study, the opposite effect of BMP4 and Noggin treatment was observed only in vitro, where addition of BMP4 protein significantly reduced the proliferation rate of cultured ENCDCs (Fig. 7F). Similar to this, the proliferative expansion of the developing ENS was restricted by the addition of recombinant BMP4 protein (Fig 4F) which opposes GDNF-induced proliferation of ENCDs as observed earlier by Chalazonitis et al., (2004, J Neurosci). As shown in Fig 7F the robust ENCDC proliferation was not influenced by the presence of Noggin, with cell proliferation up to 10%, therefore we conclude that in the absence of gut mesenchyme, Noggin has no direct inhibitory effect on ENCDC proliferation. This highlights the complexity of ENS-environment interactions and supports further studies to understand the importance of the complex microenvironment in determining susceptibility to mesenchymal-derived morphogens during ENS development.

  1. Figure 6 – not clear if F and F’ is 3C2 staining? Please explain which cells receive the virus

Thank you for the observation, to avoid any confusion we fixed the labeling.
RCAS is a replication competent avian retrovirus, which infects dividing cells.  Injection of the BMP4-RCAS resulted in infection of mesenchymal and smooth muscle compartments of the gut wall, except for the epithelium.

  1. Figure 7 E-G: consider using letter code to indicate which column is significant from which – the bars are confusing

Thank you for the suggestion, we aimed to follow the most frequently used representation method in labeling significance levels.

6)      Please refer to Fig 9 in the discussion.

Thank you for the observation, we have corrected the inconsistence.

7)      Figure 9 is hard to understand – for example, why is there an arrow going from E5 to the middle cross section of the ceca, when there are no enteric ganglia present. Is there a different color for migrating progenitor cells vs enteric ganglia? Why should the reader compare the middle-section between E5 & E7 for the midgut?

We have corrected Figure 9 based on the reviewer’s comments. The arrow going from E5 to the cross section of the ceca was removed, migrating progenitor cells were color coded black, whereas ENCC derived differentiated ENS cells are represented in green. These details were added to the figure legend.

Minor comments

-          Some of the figures use red and green color coding which is not suitable for color-blind people – please use green-magenta or other color palettes suitable for color-blind people.

Thank you for this comment. We have tried to modify the figures accordingly; switch from RGB to CYMK color or transform in grayscale, but unfortunately in most cases the figures lost the details. We are going to discuss this issue with the Technical Editor.

Reviewer 2 Report

Comments and Suggestions for Authors

This manuscript aimed to characterize the expression of BMP4 during avian hindgut ENS development and to investigate the effect of BMP signaling on ENCDC migration  and differentiation. The author  find that overexpression or inhibition of BMP4 in the ceca disrupts hindgut 23 ENS development, with GDNF playing an important regulatory role. Our results suggest that these 24 two important signaling pathways are required for normal ENCDC migration and enteric ganglion 25 formation in the developing hindgut ENS. Here are some question and comments:

First, generally we don't put any reference in abstract part.

Second, Line 23, it is "in vivo" instead of "in ovo"?

Third, How do you Inhibit of BMP4 signaling?By adding Noggin?  Why migration(increased), ganglion size(decreased) and proliferation(no change)  show different trends between inhibition and treated group?Do you try add Noggin after treated with BMP4?

Fourth, Please explain what is E5,E6,E7 mean before use them, the reader may feel confuse if they do not have the background.

Fifth,Why the BMP4 treated group and Noggin group show same trends for migration and proliferation after treated with GDNF since one is active the signal, one is inhibit the signal? 

Comments on the Quality of English Language

still need check the miss spelling errors

Author Response

First, generally we don't put any reference in abstract part.

Re: We thank the reviewer for this observation. We moved the reference to the appropriate part of the Introduction section.

Second, Line 23, it is "in vivo" instead of "in ovo"?

Re: Thank you for this comment. Corrected.

Third, How do you Inhibit of BMP4 signaling? By adding Noggin?  Why migration(increased), ganglion size (decreased) and proliferation (no change) show different trends between inhibition and treated group? Do you try add Noggin after treated with BMP4?

Re: We did not try to add Noggin after treatment with BMP4. We wanted to block the endogenous BMPs with Noggin treatment and not to confirm the inhibitory effect.

Fourth, Please explain what is E5,E6,E7 mean before use them, the reader may feel confuse if they do not have the background.

Re: Thank you for the suggestion, we added the explanation to the main text and to the figure legends (see line 107).

Fifth,Why the BMP4 treated group and Noggin group show same trends for migration and proliferation after treated with GDNF since one is active the signal, one is inhibit the signal?

Re: As shown in Fig. 4D migration changes to different directions, even though it is only a slight change, Fig. 7E also supports this finding. Proliferation rates presented in Fig. 4F suggest there should be a delicate balance in BMP levels in regard of proliferation regulation, however there is a striking difference when we only take isolated neural crest derived cell culture conditions into consideration (Fig. 7F).

Round 2

Reviewer 1 Report

Comments and Suggestions for Authors

Many of the comments have been addressed by the authors but my main comment is still regarding the boxes in the figures which do not match size wise or do not show all the channels (see details for Fig 2 and 3). This needs to be addressed.

1)      Figure 2. the box in F is still not the same size as G, G’ and G’’ – also why are the authors not showing all the channels that are depicted in F also in G-G’’ – it makes it very hard to follow what is corresponding to what if the image information (and this includes the labels in the image) doesn’t match between the separated channels of the closeups as well as between the closeup and the original image – this also applies to other figures, so please go through your figures and make sure that the close-ups match the original figure in all aspects.

2)       Figure 3. the box in A is still not the same size as A’

3)      Figure 3C: can you indicate which boxed area is depicted in C’ and C” so the reader doesn’t have to guess

4)      Figure 3 D and D’ do not match up – the image that you see in D’ is not the same as in the close-up – please make sure that this is true throughout all the figures.

5)      With regard to the use of green-red in some of the figures which is not suitable for color-blind people: switching to CYMK does not help, but changing the pseudo coloring to green – blue or magenta and green or other color palettes suitable for color-blind people is advised.

Author Response

We thank the Reviewers for their positive response and their critical comments. For better clarity the multitude of minor language and grammar corrections is unmarked, but larger changes in the text are highlighted.

Following the suggestion of Reviewer 1, we fixed the Figs. 2,3, which should make the results easier to understand. We fixed the boxes to match the size and shape of magnified insets. New figures and channels were added for Fig 2 G-G” and Fig. 3A’. We modified the image color for Figs 2,3,7,8 according to the recommendations and changed the red to magenta:

Jambor et al (2021) PLOS Biol: https://pubmed.ncbi.nlm.nih.gov/33788834/

and: https://www.nature.com/articles/s41592-023-01987-9

With this, we hope we were able to answer all questions satisfactorily and the information provided will make our manuscript acceptable for publication.

Thank you for raising this important point.

Reviewer 2 Report

Comments and Suggestions for Authors

The author answer my question.

Comments on the Quality of English Language

double check all the spelling

Author Response

We thank the Reviewer for the positive response. For better clarity the multitude of minor language and grammar corrections is unmarked, but larger changes in the text are highlighted.

Round 3

Reviewer 1 Report

Comments and Suggestions for Authors

The authors have addressed all comments.